# Re-Imagining Education for All Children

**Roy McConkey** [1,*] and **Judith McKenzie** [2]

1. Institute of Nursing and Health Research, Ulster University, Belfast BT15 1AP, UK
2. Division of Disability Studies, Department of Health and Rehabilitation Sciences, Faculty of Health Sciences, University of Cape Town, Cape Town 7701, South Africa; judith.mckenzie@uct.ac.za
* Correspondence: r.mcconkey@ulster.ac.uk

**Abstract:** Universal education is an elusive goal in many countries, especially for disabled children. Nonetheless, determined efforts around the globe have shown that it can become a reality once existing systems were re-imagined by practitioners who arguably have been to the fore more so than academic researchers. Their efforts have identified new ways of thinking about children's disabilities, the introduction of new practices in schools, forging partnerships between teachers and parents and mobilising community resources. Societal change is both a consequence of and a support to these local systems. The complexity of creating education for all may be daunting, but it is achievable when driven by committed, creative and inspirational leadership from practitioners, as is evident from the examples provided in this paper, which were further validated by research and evaluation into their efforts.

**Keywords:** disabled children; learners; inclusion; rights; parents; schools; community; society





## 1. Introduction

In the course of human history, national educational systems are a new phenomenon. Ireland was one of the first countries in the world to establish a national school system under the British colonial administration. It began in the 1830s and made education free for children, with the government paying for school buildings and the salary of teachers. In due course, similar initiatives have occurred in every nation, but the task of developing a universal education system for all children is incomplete in most, if not all, countries across all continents [1].

Whether intentionally or not, around the world many children have been—and some still are—denied an education, be it on the basis of gender, age, religion, ethnicity, social class or ability/disability. Some children are at even greater risk of exclusion if they experience multiple disadvantages, for example, girls from ethnic minorities with disabilities. However, poverty remains the greatest exclusionary factor across all children and this is exacerbated in situations of disaster or forced migration [2].

Over the past 35 years, we, the authors of this paper, have been fortunate to work with local practitioners in many countries around the globe—Africa, Asia, Eastern Europe and the Caribbean—who were striving to make education available to all children, but especially those with physical, sensorial, social and intellectual impairments. We refer to them as 'disabled children' in recognition that their disabilities arise not just from bodily impairments, but that social and environmental factors also play a considerable part in their exclusion from education and wider society.

In many countries, disabled children are enrolled in special schools instead of attending the same school as their non-disabled peers. But this option is mainly confined to urban settings and does not meet the needs of the population nation-wide. Consequently, many disabled children are out of school and children with intellectual impairments are those most likely to suffer exclusion. International experience now recognises that the creation of a parallel, segregated system of schooling for disabled children is inefficient and ineffective,

particularly in making education available to all such children irrespective of their country of residence [3].

This article shares our personal reflections on how inclusive education can become a reality even in the most impoverished communities. Sadly, our mentors are too many to mention by name, but should they read these words, they will recognise their contribution, and for this we are very grateful. We have also been fortunate to draw on the experience and expertise of an international collaborative of researchers and practitioners convened by the Special Olympics Global Center for Inclusion in Education (https://www.specialolympics.org/what-we-do/youth-and-schools/global-center-for-inclusion-in-education?locale=en, accessed on 10 September 2023).

At the forefront of our mind was the sharing of these insights with other practitioners—educationalists, therapists, policy makers, parents and disability activists; so, in this article, we deliberately use illustrations alongside accessible, rather than academic, language. Nevertheless, we recognise the need for more research, especially in identifying how the 're-imaginings' can be implemented in even the least resourced settings. Hopefully, this paper scopes the width and depth of the research and evaluation endeavours that are needed in the coming years.

## 2. Why Inclusion?

Inclusive education is built on three key principles: rights, benefits and universality.

- Rights

First, ALL children have the right to education. It was only in 1989 that this right was formalised by the United Nations in the Convention of the Rights of the Child [4]. Article 28 states:

*"States Parties recognize the right of the child to education, and with a view to achieving this right progressively and on the basis of equal opportunity, they shall, in particular: (a) Make primary education compulsory and available free to all; (b) Encourage the development of different forms of secondary education . . . (c) Make higher education accessible to all."*

In many countries, these rights and others contained in the UN Convention of Rights for Persons with Disabilities [5] were already incorporated into national policies and laws or have been since the UN Conventions were signed. In South Africa, for example, Section 29(1) of the Constitution [6] states:

*"Everyone has the right—(a) to a basic education, including adult basic education; and (b) to further education, which the state, through reasonable measures, must make progressively available and accessible."*

- Inclusion benefits everyone

Educating children with disabilities helps teachers to gain extra skills and expertise that will help all their pupils and not just those with disabilities. Other children benefit socially as they learn appropriate ways of dealing with difference. The wider society benefit as children with disabilities are enabled to become productive members in their family and community [7]. As the UNESCO policy guidelines noted:

*"Inclusive schools are able to change attitudes toward diversity by educating all children together, and form the basis for a just and non-discriminatory society."* [8]

- Universality requires changes to current systems

The move to universal education for all children is incomplete in all countries, with some further from this goal than others [1]. Some countries have invested heavily in special schools, but even in mainstream schools, current systems with their ways of thinking and working were not designed to be inclusive of children with different abilities and from different social backgrounds [9]. Hence, the General Comment on UN Convention on the Rights of persons with disabilities asserted:

> "*The right to inclusive education encompasses a transformation in culture, policy and practice in all educational environments to accommodate the differing requirements and identities of individual students, together with a commitment to remove the barriers that impede that possibility.*" [10]

And as Albert Einstein astutely observed:

> "*We cannot solve our problems with the same thinking we used when we created them.*"

## 3. Changing Educational Systems

Figure 1 illustrates the different systems involved in education, which is taken from Bronfenbrenner's socio-ecological model of human development [11]. The model is built around the key influences on a child's development. Over the past three decades, our understanding of these influences have changed greatly. No longer is the focus on the child's biology, which has dominated our thinking especially when children are born with or acquire impairments of the body, senses and brain. This thinking resulted in what has been termed a 'medical' model of disability, with a focus on correcting or remediating the children's impairments.

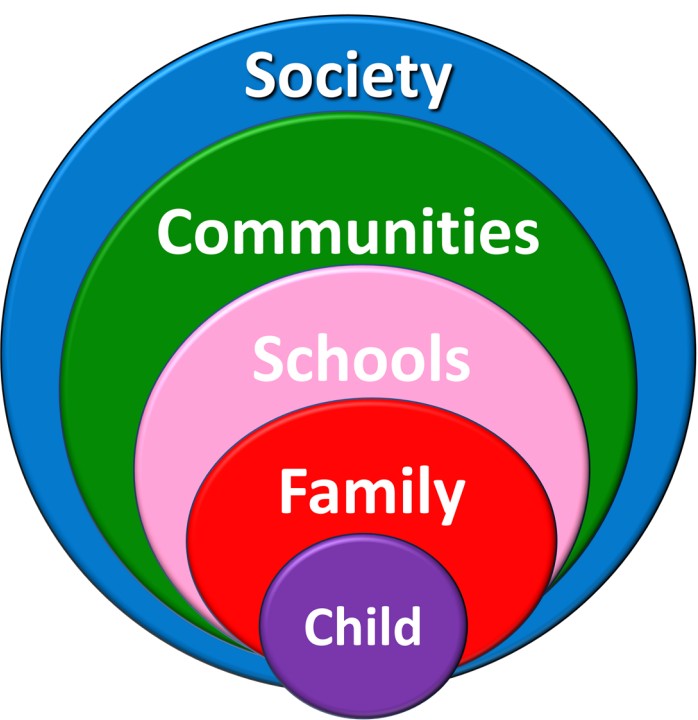

**Figure 1.** The systems involved in educating a child.

In recent years, however, the biological emphasis has shifted to focusing on the social, psychological and environmental influences on all children's development. Indeed, this shift, embodied in the World Health Organisation's International Classification of Functioning [12], is far more hopeful for disabled children as educational and social interventions can ameliorate the impact of bodily impairments, thereby enabling children to participate more in the life of their family and community. Attention moves toward the learning environment and how barriers that prevent disabled children from fully participating can be progressively removed. Furthermore, the priority becomes creating a good quality of life for the child rather than on 'fixing' his or her impairments [13].

Figure 1 illustrates the other systems in human society that influence the development of all children. They too are equally, if not more, important for children at the risk of exclusion from education. The remainder of this paper will focus on the changes required to make these systems more suited to the inclusion of children with disabilities, but they likely contribute to making universal education a reality for all children.

## 4. Re-Thinking Education

A common mistake is to think that education is the preserve of schools. Worse still is to consider that academic achievements—perceived to be the root to economic advancement—are the main outcomes from schools and even education. Such thinking has reinforced the exclusion of students who in the past were deemed 'ineducable' and then denied admission to their local schools. Instead, they may have attended special centres or, more likely, remained at home isolated from their non-disabled peers.

Rather, the African proverb—*it takes a village to educate a child*—provides a different perspective. The UN Committee responsible for overseeing the UN Convention on the Rights of the Child presented a vision of education that extends beyond schools [4].

"*Education must include not only literacy and numeracy but also life skills such as the ability to make well-balanced decisions; to resolve conflicts in a non-violent manner; and to develop a healthy lifestyle, good social relationships and responsibility, critical thinking, creative talents, and other abilities which give children the tools needed to pursue their options in life.*"

Indeed, when we look back at our own childhood, we received so much more from attending school than sitting in classrooms learning to read, write and pass examinations. We learnt how to get on with other children and to make friends; our physical and emotional wellbeing was nurtured through sports, for example; we were exposed to the culture of our community and we were prepared for the world of work. Research on children's education during the COVID-19 epidemic found it had a devastating impact on the children's social and emotional development, and not only on academic attainments [14]. So, refusing children a place in schools because they struggle to pass exams denies them the experiences needed to become an active and productive citizen in their community and country.

## 5. The Crucial Role of Schools

Nevertheless, schools are central to advancing education for everyone and are established and funded in all cultures globally. But what has changed over time is who schools aim to educate and what they aim to achieve. We should also remember that 'schools' exist in many different manifestations and for learners of all ages—preschool, primary, secondary, grammar, technical, colleges and universities. Making these diverse 'schools' inclusive for all their students is still a work in progress internationally.

Over the past four decades, there have been concerted efforts globally to make schools more welcoming of all learners, especially those who might previously have been excluded. The schools may range from preschools to universities, but most attention has been paid to primary education. Hence, at present, we have a reservoir of knowledge to draw on, in order to make schools more inclusive [15].

In recent years, the members of the United Nations have agreed on 17 sustainable development goals (SDG), of which Goal number 4 commits them to "*ensuring an inclusive and equitable quality education and promoting lifelong learning opportunities for all.*" [16]. The SDG is explicit that 'all means all' and the Global Education monitoring report of progress up to 2020 makes it clear that those who are the most excluded should be prioritised in all efforts toward this goal.

Figure 2 summarises five strategies that have been validated internationally for advancing inclusive education within schools. These have been actioned, for the most part, by committed head teachers and teachers in their own schools, often with inadequate resources, at least initially. They were motivated by their hearts but lead by their heads as they found out what worked and what did not. As Mandela proclaimed: "*A good head and a good heart are a formidable combination.*"

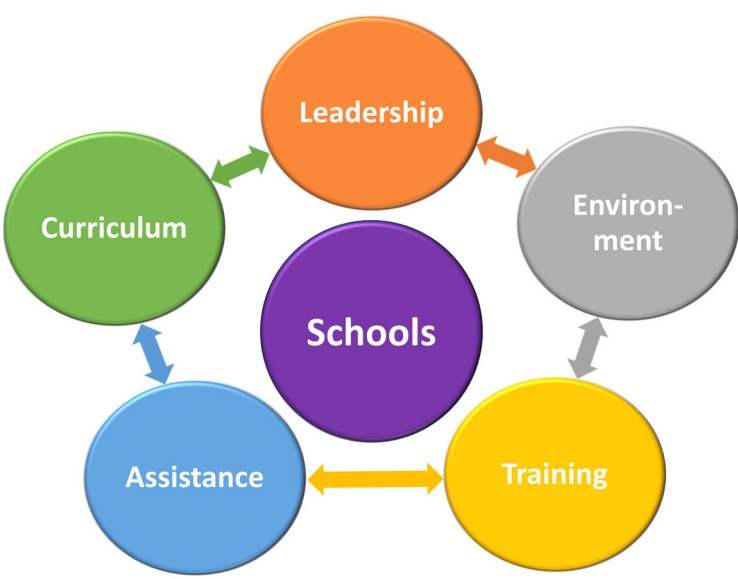

**Figure 2.** Preparing school systems for inclusion.

- Leadership

Leadership is essential for instigating and sustaining a policy of education for all children. First and foremost, this comes from the leadership of head teachers alongside other senior staff within the school. Likewise, the management boards of schools need to embrace and advocate for a policy of inclusion. But, in our experience, the most commonly given reason for why schools adopt an inclusive approach is because the leaders believe it is every child's right to attend their local school and that schools can and should make it happen. Their actions start from the heart and the belief that exclusion on the grounds of disability is equivalent to exclusion based on, for example, gender or race.

- Environment

Practicalities soon take over. Making adjustments to the school environment are essential to making all children feel welcome at school. This may include the provision of ramps and improved toileting facilities for learners with mobility difficulties, improving classroom layouts, acoustics and assistive devices for learners with sensory difficulties and having learning aids available that are adapted to learner's cognitive abilities. It also means recognising the diversity amongst students and adapting to their language, gender or racial differences, amongst others. These environmental changes benefit all students as well as teachers and often can be implemented with limited finances.

- Training

The confidence of teachers is boosted, and their attitudes changed through the provision of training in adapting their classrooms and teaching strategies to meet the needs of learners with additional difficulties [17]. Such training can come from handbooks, in-school training sessions and online courses, but often one-to-one support from colleagues and visiting experts, such as psychologists and therapists, is especially helpful to assist teachers to cope with the particular needs of individual children. Equally, teachers can build a 'community of practice' within their school that provides them with ongoing support as they share with one another how they have met the challenges they face in teaching pupils who require an adapted curriculum.

- Assistance

It cannot be left to teachers to conduct all the work of assisting learners with special needs. Various forms of assistance have evolved. More able pupils can be used as peer mentors to assist with less able children [18]. Schools have appointed a member of staff as a 'Special Needs Co-ordinator' who receives extra training so that she can provide advice

and guidance to other teachers. A teaching assistant can be allocated to work in classes with teachers to ensure learners obtain the extra support they may need. However, care must be taken that the assistant does not become a barrier to learning by allowing teachers to shift their responsibility for teaching the disabled child onto the assistant. One way of preventing this is by attaching the assistant to the classroom as a support for all the pupils and not to one learner.

Although these teaching assistants are paid in affluent countries, their role can be taken on, in part, by volunteer helpers, such as grandparents or retired teachers. Indeed, recruiting, training and paying for additional assistance from existing community members also strengthens the learners' inclusion beyond the school. Hence, school budgets should make provision for new forms of staffing.

- Curriculum

Arguably the most significant step towards making schools inclusive are the adaptations that teachers make to the curriculum they use with their learners. These adaptations include adjusting the level of difficulty of the subjects they teach, replacing certain subjects with extended time by other subjects and adjusting their teaching strategies, such as making more use of visual materials and finding other means for assessing the children's learning, rather than examinations. The Universal Design for Learning provides a valuable road map for schools [19], as it is a framework for providing multiple pathways to learning through adaptation of the ways in which information is presented, how students engage with this in their learning and how their learning can be expressed and measured or monitored. Schools may also place more emphasis on non-academic activities—such as sports, music, art, cooking and farming—with learners who are less able to cope with academic subjects.

Although training courses can help teachers to make adaptations to the curriculum, it is often the creativity and expertise of the individual teachers that makes sure they are put into practice.

- Monitoring progress

Underpinning all these efforts is the ethos of schools regularly monitoring their progress towards making their schools more inclusive by identifying what works and what they need to change to conduct a better job. This should be a process that is supportive of teachers, empowering them to continuously improve their practice and addressing valid concerns which they might have that stand in the way of building inclusive classes. The self-reflections of school leaders and staff can be supported by newsletters to parents, reports to the Board of Management and visits from school inspectors to reinforce their contribution in sustaining an inclusive ethos within the school [20].

## 6. Preparing Children

Inclusion in education needs to be planned long in advance of the child starting school. Indeed, the earlier it begins in the child's life, the more likely successful inclusion in a mainstream school will be. Much of this preparation will fall on families, but most of them require advice and encouragement from others in the locality. The following are some examples that are summarised in Figure 3 and that draw upon the WHO's Nurturing Care Framework for Early Child Development [21].

- Screening for impairments

Parents often harbour concerns about their child's development but are reluctant to share them with other family members, who in any case may offer unfounded reassurance that nothing is amiss. Conversely, parents may resist the suggestion from others that the child may have difficulties. In many countries, the solution is to offer development screening alongside child development checks in local clinics [22]. This can be performed as part of immunisation or augmented feeding programs, which incidentally are even more important for infants with developmental problems that are evident at birth. Checks on children's vision and hearing can be undertaken and delays in meeting developmental

milestones can be confirmed. Referrals can be made to health professionals—where they are available—for the provision of assistive aids, such as glasses, hearing aids or mobility aids.

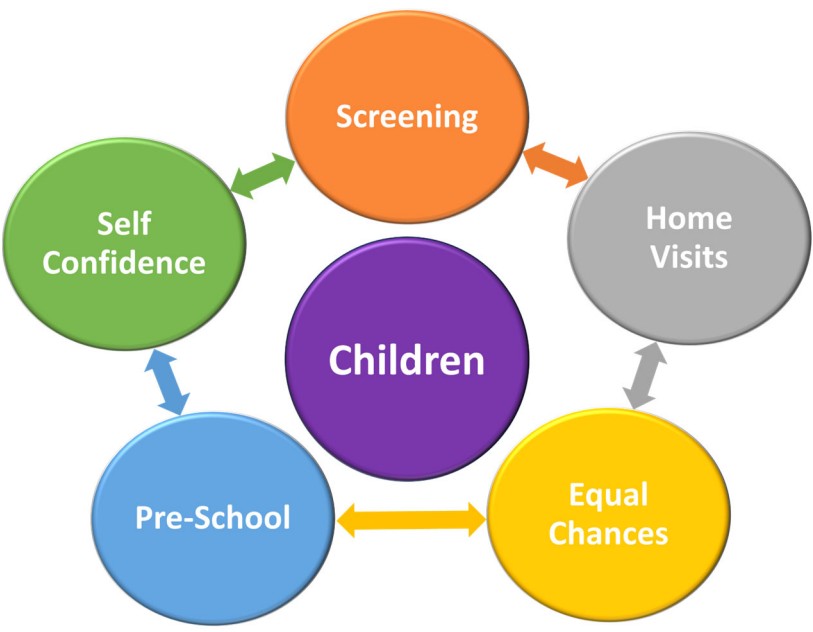

**Figure 3.** Preparing children for inclusion.

- Home-based interventions

    There is convincing evidence of the effectiveness of home-visiting schemes to advise and engage families in promoting their child's development in the preschool years [23]. These may be provided by community health staff, such as nurses, community-based rehabilitation workers or therapists. The goal is to boost the developmental progress of infants and toddlers using everyday activities and equipment, while showing families how they can assist the child to overcome their difficulties.

- Equal chances

    Young children with developmental delays deserve to be treated the same as others in the family. Parents who embrace this ethos become advocates for inclusion within the wider family, their neighbourhood and ultimately the wider community simply by ensuring the child participates in family and community events, such as parties, religious celebrations and even routines, such as visits to the market. All of this challenges the stigma and shame that families have experienced in many societies around the world [24].

- Preschool education

    Prior to entering school, children benefit from attending preschool facilities, such as playgroups and creches, and then early education centres, such as kindergarten or nursery schools [25]. These provide opportunities for children to learn new skills, to interact with other children and to be socialised into appropriate behaviours within classrooms. It also provides parents with opportunities to obtain further advice from the educators as well as meeting other parents. Where such facilities do not exist, parents with support from community volunteers have instigated such centres.

- Boosting the child's self-confidence

    All of the foregoing helps to build the child's self-confidence and their self-esteem, which in the past did not happen when they were hidden away in the family home. Ironically, this attitude was perpetuated if they were sent to special schools. The early experience of inclusion also increases the young child's chances of fitting more easily into the school context as they may know their fellow classmates and have experience of how to behave in classrooms.

To date, the preschool experiences that are a prelude to school enrolment have received little attention. Hopefully, this will be rectified in the immediate future, but it requires education, health and social services to work together in a more collaborative manner, which admittedly has been an elusive ambition to make happen [26].

## 7. Preparing Families

Parents are "*the main and natural educators of children*"; thus states the Constitution of the Republic of Ireland (1937). In many countries, it is not only the parents but also the extended family and even other community members who take on this role. In South Africa, for example, grandmothers are often the caregivers, especially in rural environments. The educational role of families has been borne out by subsequent research studies and is no less true when children have special needs arising from a disability [27].

However, family members can feel at a loss to know what to do for the best, and their feeling of helplessness is compounded if they experience negative attitudes within the wider family circle about disability. Yet, their active involvement in the child's education is essential to complement the efforts of schools. Hence, schools that aspire to be inclusive make great efforts to support parents. Figure 4 shows five ways in which teachers have successfully partnered with families.

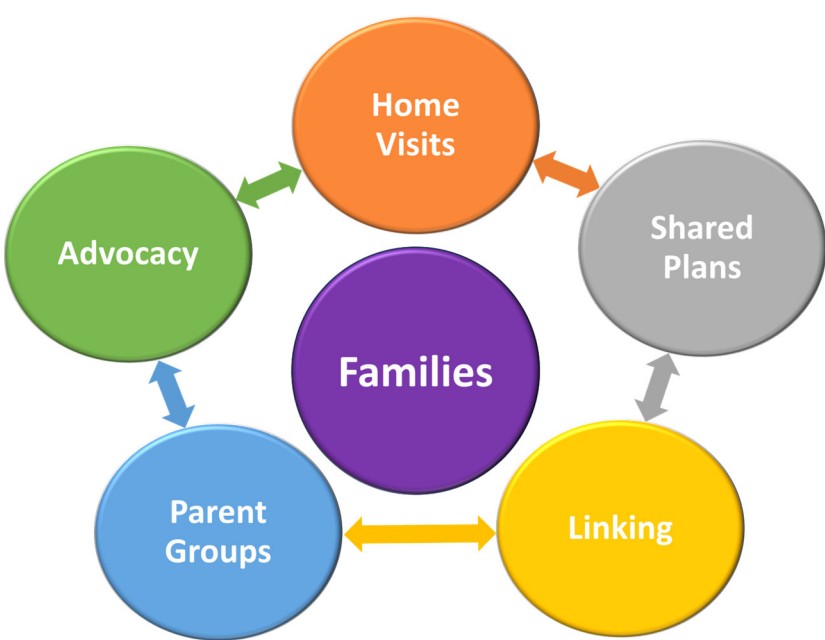

**Figure 4.** Preparing families for inclusion.

- Home Visits

Teachers are provided the time away from school to visit the family at home so that they can see the child's home environment and meet the significant people in the child's life [23]. The visits help to build a personal and trusted relationship between the teacher and families in ways that inviting the parents to the school may fail to do. Such visits are especially important in the early years when the child starts school. They may occur monthly and, in time, visits can be replaced by telephone chats or the exchange of home-school diaries.

- Shared plans

Teachers and parents work together in devising an Individual Education Plan for the child with special needs. Teachers need to understand family priorities and needs in determining which goals required to be addressed, deciding with families what should be included in the Plan and how they will be addressed. These plans are a legal requirement in the USA and the UK. They outline the activities that will be used in the home as well as

in school to achieve the child's goals. A formal review of the plan is held each year in the school to which parents are invited, but informal reviews can occur more regularly between teachers and parents so that plans can be adjusted according to the child's progress and emerging needs [27].

- Linking

In recent years, these plans have evolved into Child and Family plans, in recognition of the unmet needs of family members that may impact on the child. Although schools are not expected to meet these needs, teachers can signpost families to where help might be available. This might take the form of providing parents written details of people to contact or it could extend to having someone from the school accompanying the parent to their first appointment. These links can be strengthened as schools build relationships with other support services in the local community; we will come back to this point in the next section.

- Parent Associations

Parents can gain knowledge and practical and emotional support from meeting with other parents who have children similar to theirs. Some schools provide a room for parents to meet one another after they drop the children off at school or before picking them up. Schools may hold information events at which parents and other family members can socialise as well as becoming better informed on certain topics. Families can be supported to form their own association. These exist in many countries for specific conditions, such as Down syndrome or autism.

- Advocacy

A crucial contribution of parent associations is bringing family needs and those of the children to the attention of health and social services and government agencies. Their advocacy has been a potent influence in the development of supports for children with disabilities in many countries of the world [28]. Parents can speak with greater authority and authenticity than professionals (although their support is very welcome), which is why, in many developing countries, parent associations are open to other family members as well as professional friends.

Of course, parents vary in the extent to which they are able or have the interest or resources to become involved in school and community activities. But even the efforts of a small group of willing parents can make a considerable difference in opening up opportunities for greater cooperation between schools and families.

## 8. Preparing Communities

The third system that needs to change is enhancing the contribution that local communities can make to advancing inclusive education. The African proverb "*It takes a village to educate a child*" acknowledges the truth that families and schools alone are not up to the challenge of preparing learners with additional needs to become valued and productive members of their local community.

Yet, this relationship between schools and the communities they serve has received little attention and, indeed, in more affluent countries and in cities they seem to have grown apart. Arguably, the opportunities for linking schools and communities are greater in smaller towns and villages and these experiences can extend into cities and their neighbourhoods [29]. Figure 5 summarises five means of building links between school and communities.

- Leaders

Every community has its formal and informal leaders, be they local politicians, religious leaders, traditional elders or business owners. Schools and families can enlist the support of leaders in advancing opportunities for learners with special needs. For example, the community leaders could support fund-raising efforts for the equipment or adaptations of school premises. They could create work experience opportunities for school leavers

that might lead on to paid employment. In our experience, many community leaders have a personal experience of disability within their families and welcome the chance to become more involved in creating better communities for everyone.

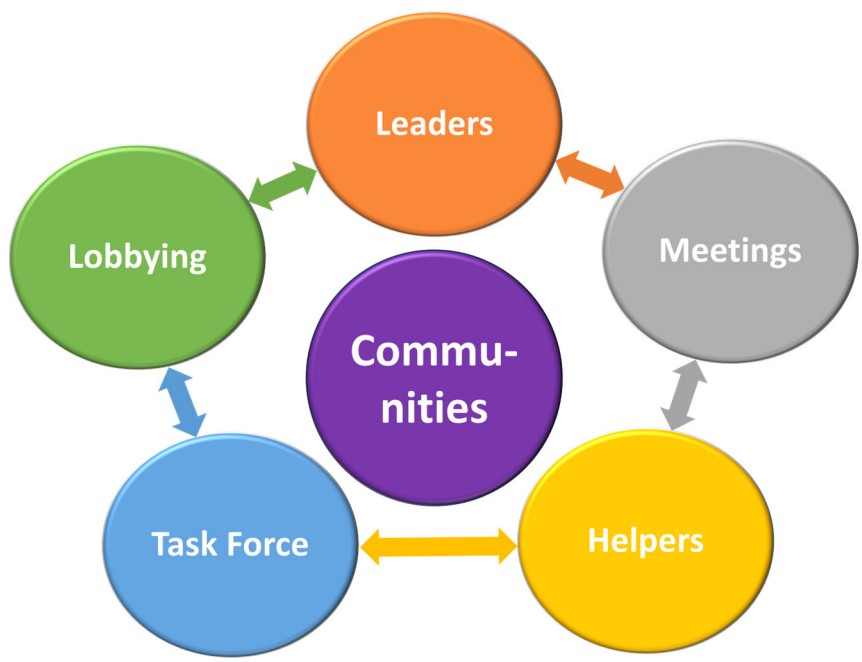

**Figure 5.** Preparing communities for inclusion.

- Meetings

An effective strategy for informing the wider community is by holding meetings that are open to everyone. In rural Africa, these have taken the form of village gatherings at which songs, dances and short dramas intermingle with speeches from teachers and parents about disability and breaking down the myths and taboos that surround these conditions. Examples are provided as to how the community can help in terms of fundraising but more importantly of 'friend raising'.

- Helpers

Volunteers can be recruited from local communities to assist with projects in schools, such as assisting with reading lessons or in after-school activities like sports. Families too can receive assistance through donations of food and clothing but also by undertaking some child-minding when parents have to attend appointments, for example. Or volunteers may also be able to help families who require transport. In affluent countries, mentoring schemes have been developed in which non-disabled youths are paired with a special needs student on work experience placements or for leisure activities. For example, Special Olympics is an international sports organisation that has successfully recruited volunteers as sports coaches, players and helpers.

- Task Force

Community representatives can be invited to join a task force for the specific projects that schools aim to undertake. Examples include improving the accessibility of school buildings and facilities, obtaining assistive devices for learners to use in the classroom and developing income-generation activities for school-leavers. Community members often have social and business contacts they can approach for the help that teachers and parents may lack.

- Advocacy

Community support can be vital when it comes to lobbying government and service agencies for increased resources to meet the needs of learners, families and schools. This

may start with short-term, small-scale requests but can grow into lobbying for medium- and large-scale changes to policy as well as practices that have an impact beyond schools and the immediate community.

## 9. Changing Society

In previous sections, our focus was on the local actions undertaken by local people using local resources. Indeed, inclusion has to start from where people are—in families, schools and communities. We can take heart from Margaret Mead's assertion that large numbers of people are not a requirement; rather, she claimed:

> "*Never doubt that a small group of committed citizens can change the world. Indeed it is the only thing that ever has.*"

Even so, local initiatives must also have an eye to the wider horizon of bringing about societal shifts from exclusion to inclusion. Figure 6 captures some of the strategies for making this happen.

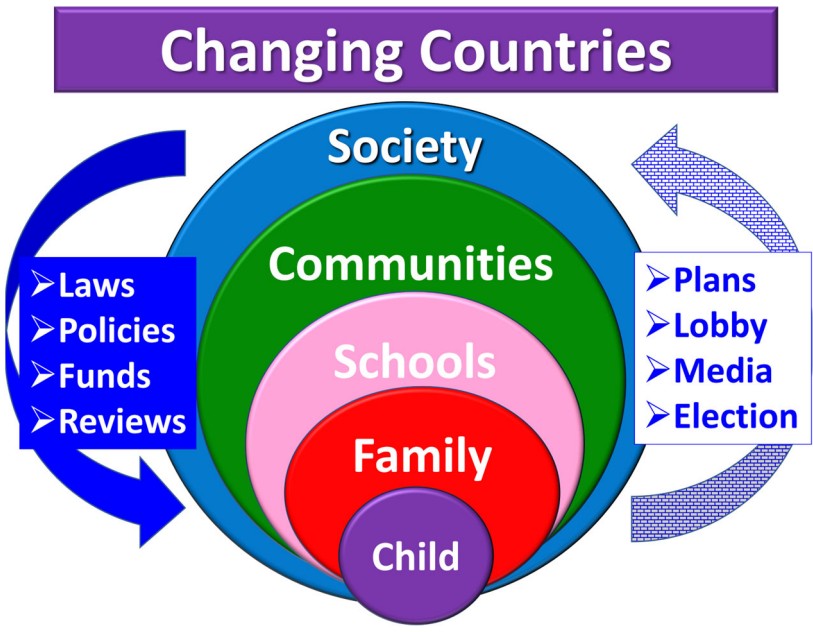

**Figure 6.** Making nations more inclusive.

Two forces need to be mobilised and ideally each supports the other. On the right are the ones that could be described as bottom-up approaches as they emerge from the actions of communities, schools and families in order to influence the wider society and notably government policy and action [30]. Those on the left are those that come from the top down with the aim of strengthening and supporting communities, schools, families and children.

- Bottom-up approaches

The strategies here include the preparation of plans and proposals for the improvements needed to advance inclusive education in the nation. Meetings with key officials and politicians can be sought to lobby for the changes needed while emphasizing to them the benefits to be obtained as well as pointing out the consequences of inaction. Enlisting the help of the media—TV, radio and print—is another fruitful strategy for bringing your concerns to the attention of wider society. Election time in democratic societies is a valuable opportunity for gaining the support of political parties and lobbying to have your proposals included in their election manifestos.

- Top-down approaches

Governments can shape society in many ways; indeed, it is one of their primary functions. The strategies commonly used include the formulation of laws that require

actions to be taken by organisations, systems and individuals with sanctions for those who break the law. Statements of policy are a further means by which society can be guided and the actions that are expected to be taken by people charged with implementing the policy. The allocation of funds in support of laws and policies is vital, although the monies often fall short of what is needed and sometimes they never materialise. Undertaking reviews of policies at regular intervals is a further means for identifying achievements but also the shortcomings and how they might be addressed. Increasingly, reviews of national policy focus on the value for money that existing services provide and how a greater value can be obtained from the monies allocated as well as judging how new funds could be better used.

At best, these two forces will dance in synchrony, with each supporting the other so that change actually does happen. At worse, they act alone or not at all. The efforts of those coming from the bottom up may be ill-defined and fragmented; thus, they are easily rebuffed by those at the top. Likewise, many well-intentioned, top-down initiatives wither when communities, schools and families are ill-prepared, often through resources that are not provided.

## 10. Conclusions

Universal education for all children is a complex endeavour and the present void between policy and practice needs urgent attention. Practitioners have shown what can and needs to be done, but to date, their efforts have been disparate and uncoordinated, nationally and internationally. The priority is to gain a better understanding of how inclusive education can be implemented within and across the different domains identified in this paper. We recognise too that some of the strategies we outlined will not be feasible to implement in all countries due to varying cultural, social and economic circumstances. Nonetheless, research will be vital into the processes involved in advancing educational inclusion national and internationally, as well as documenting the outcomes attained. The insights gained from proponents of what is known as 'implementation science' should provide useful guidance for future research and evaluation studies into inclusive education, especially in less resourced countries [31].

Above all though, we must not lose sight of the noble aims that drive education for all the world's children. As Kofi Annan, a former secretary general of the United Nations, concluded: *"Education is a human right with immense power to transform. On its foundation rest the cornerstones of freedom, democracy and sustainable human development."* Sadly, Goal 4 of the SDGs that was to be attained by 2030, namely, to *"ensure inclusive and equitable quality education and promote lifelong learning opportunities for all"*, is unlikely to be met internationally [16].

Universality is perhaps an unattainable educational dream and may not be achieved in the coming seven years. Yet, we know the strategies that can make it possible, but these require a great deal of effort, creativity and commitment on the part of many players, all of whom need to share a common goal and arguably sacrifice some their comforts and advantages for the common good. Hence, we must expect progress to be slow and at times fractious, disheartening and frustrating. We are defeated if we let these setbacks deter our quest. Rather, they must fuel our vision and commitment and, in the words of the Irish poet Seamus Heaney, *"believe that a distant shore is reachable from here"*. We all have a part to play. What can and should you do?

**Author Contributions:** R.M. and J.M. contributed equally to all aspects of the review. All authors have read and agreed to the published version of the manuscript.

**Funding:** This research received no external funding.

**Institutional Review Board Statement:** Not applicable.

**Informed Consent Statement:** Not applicable.

**Data Availability Statement:** Not applicable.

**Acknowledgments:** Our thanks to Florian Kiuppis, Kate Lapham and Jane Wairimu for their helpful comments on an earlier draft of this paper.

**Conflicts of Interest:** The authors declare no conflict of interest.

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
