# Peer review of "Re-Imagining Education for All Children"

_disabilities, doi:10.3390/disabilities3040033_

Round 1

Reviewer 1 Report

Comments and Suggestions for Authors

The proposed contribution is not clearly academic (it rather sounds like a popularization article or "a sermon"...), and we can can find there even some significant simplifications, but I appreciate the practical sound and practical suggestions at the same time. Undoubtedly, the article is clear enough for a wide range of readers even from non academic fields...

A lot of elementary, well known and already (frequently) published information and arguments about inclusive education are repeated here. In spite of that, the authors formulate and remind some extremely important ideas regarding the right for education for all as well, and they also point to the edcuational perspectives that exceeding heavily rooted myths on the real meaning and content of education of human being today.

There are mentioned some important aspects of so called inclusive education not only from the perspective focused on pupils with disabilities, but also with other kinds of exceptionalities. However, I miss the wider perspective of inclusion as a specific concept of social coexistence where the education itself is only a small (but very important) part of it. Let's also draw attention to some arguable opinions and ideas - e.g. that the inclusive changes in a wider society grow from the local examples of good practice; It doesn' seem to work this way, but it seems that a focus on changing the paradigm of thought in society (not just changing some basic laws and policies) is necessary for the successful implementation and promotion of truly inclusive practice in local schools and communities...

What I really appreciate is not simply ctitical and negative tune but focus on some positive and constructive suggestions and challenges in the topic. On the other hand, the authors should be more attentive to significant culture differences in countries around the world; it must be stated that many suggestions are not applicable everywhere. I also miss the clear "People First Language" (as an important inclusive element) in some part of the text (e.g. "children with disabilities" is better thant "disabled children" etc...).

Anyway, such a theoretical paper which can lead readers to deeper rethinking conceptual connections of inclusion (not only in education) can be very valuable contribution among many of strict research studies...

Reviewer 2 Report

Comments and Suggestions for Authors

This article provides a helpful summary of the work that can be done globally to provide inclusive schools and settings.  The authors extensive experience and knowledge is clearly helpful in making this an accessible and informative article.  In many cases, the focus is about resources which are needed, but this article takes a more pragmatic approach.  By highlighting actions that can be conducted across a range of stakeholders, this makes for interesting reading.  Although I am in overall support of this submission, I’m making the following recommendations:

The use of the literature is somewhat limited.  Points are made with very little use of the literature (see for example page 7, although this is not the only place).  The authors make the case for a less academic writing style which is fine, but the use of the literature to back up the reflective writing is still necessary. 

I’m querying the point made (line 31) about poverty being the ‘greatest exclusionary factor…’.  I do agree of the importance of this, but the citation used links to the 2020 Global Education Monitoring Report Summary.  When I look at this document it seems to indicate that disability is a greater consideration.  If the authors feel that I’ve misinterpreted their point, perhaps a direct quote would be helpful here instead. 

Overall this is a useful summary and worthy of publication. 

Author Response

This article provides a helpful summary of the work that can be done globally to provide inclusive schools and settings.  The authors extensive experience and knowledge is clearly helpful in making this an accessible and informative article.  In many cases, the focus is about resources which are needed, but this article takes a more pragmatic approach.  By highlighting actions that can be conducted across a range of stakeholders, this makes for interesting reading.  

Many thanks for this affirmation.

Although I am in overall support of this submission, I’m making the following recommendations:

The use of the literature is somewhat limited.  Points are made with very little use of the literature (see for example page 7, although this is not the only place).  The authors make the case for a less academic writing style which is fine, but the use of the literature to back up the reflective writing is still necessary. 

As you will appreciate, there is an extensive literature we could have drawn upon and the 28 citations we had given seemed to us to be the most pertinent to substantiate the points we have made.  Moreover, certain references could come in more than one section and we have now made this clearer by re-citing articles.  Plus we have added some additional references on page 7 and following, making 31 citations in all.

I’m querying the point made (line 31) about poverty being the ‘greatest exclusionary factor…’.  I do agree of the importance of this, but the citation used links to the 2020 Global Education Monitoring Report Summary.  When I look at this document it seems to indicate that disability is a greater consideration.  If the authors feel that I’ve misinterpreted their point, perhaps a direct quote would be helpful here instead. 

We have clarified in line 31 that when it comes to the exclusion of all children (and not just those with disabilities, that poverty is the major and common reason for exclusion. 

Overall this is a useful summary and worthy of publication. 

We very much appreciate the time and thought you have put into reading the paper.